# Predicting Response to Radical Chemoradiotherapy with Circulating HPV DNA (cHPV-DNA) in Locally Advanced Uterine Cervix Cancer

**DOI:** 10.3390/cancers15051387

**Published:** 2023-02-22

**Authors:** Susan Lalondrelle, Jen Lee, Rosalind J. Cutts, Isaac Garcia Murillas, Nik Matthews, Nicholas Turner, Kevin Harrington, Katherine Vroobel, Emily Moretti, Shreerang A. Bhide

**Affiliations:** 1The Institute of Cancer Research, Fulham Road, London SW3 6JB, UK; 2The Royal Marsden Hospital, Fulham Road, London SW3 6JJ, UK; 3The Royal Marsden Hospital, Downs Road, Sutton SM2 5PT, UK; 4Imperial College London, South Kengsington Campus, SW7 2AZ, UK

**Keywords:** cervical cancer, plasma HPV DNA, response prediction, circulating DNA, next generation sequencing

## Abstract

**Simple Summary:**

Cervix cancer is largely (95%) caused by human papilloma virus (HPV). Curative treatment for locally advanced cervical cancer (progressed beyond the cervix) is chemotherapy and radiotherapy (CRT), followed by internal radiation (brachytherapy). In approximately 30% of patients the cancer will return, usually within 2 years of completing initial treatment. Assessment after CRT is by clinical examination and MRI and PET/CT scans. None of these methods are particularly sensitive for detecting early relapse or discriminating between early relapse and treatment related scarring. Currently, there is no test that can reliably predict early relapse. We have developed a blood test which measures the presence of HPV-DNA fragments in the blood. In this initial feasibility study, we have demonstrated that cHPV-DNA can be accurately detected in the blood before CRT and if present at the end of treatment or detected during follow up, is indicative of tumour relapse.

**Abstract:**

Background: The majority of locally advanced cervical cancers (LaCC) are causally related to HPV. We sought to investigate the utility of an ultra-sensitive HPV-DNA next generation sequencing (NGS) assay—panHPV-detect—in LaCC treated with chemoradiotherapy, as a marker of treatment response and persistent disease. Method: Serial blood samples were collected from 22 patients with LaCC before, during and after chemoradiation. The presence of circulating HPV-DNA was correlated with clinical and radiological outcomes. Results: The panHPV-detect test demonstrated a sensitivity and specificity of 88% (95% CI-70–99%) and 100% (95% CI-30–100%), respectively, and correctly identified the HPV-subtype (16, 18, 45, 58). After a median follow up of 16 months, and three relapses all had detectable cHPV-DNA at 3 months post-CRT despite complete response on imaging. Another four patients with radiological partial or equivocal response and undetectable cHPV-DNA at the 3-month time point did not go on to develop relapse. All patients with radiological CR and undetectable cHPV-DNA at 3-months remained disease free. Conclusions: These results demonstrate that the panHPV-detect test shows high sensitivity and specificity for detecting cHPV-DNA in plasma. The test has potential applications in assessment of the response to CRT and in monitoring for relapse, and these initial findings warrant validation in a larger cohort.

## 1. Background

Locally advanced cervical cancer (LaCC) FIGO (2019) stage IB3-IVA is treated with external beam radiotherapy (EBRT) and concomitant cisplatin, followed by image guided adaptive brachytherapy (IGABT). In contemporary series this leads to excellent 5-year local control rates of 89%, and 5-year overall survival of 65% across all stages [1]. When relapse occurs, it is most often metastatic but can include cases of salvageable para-aortic lymph node relapse or oligometastatic disease. A smaller proportion of patients will also experience persistent or recurrent disease within the cervix or uterus, amenable to salvage surgery [2]. The identification of persistent disease and early relapse is challenging and largely relies on multiparametric functional imaging for which differentiating between disease and treatment related changes is difficult [3,4]. Often, recurrent or persistent disease may not be detected radiologically until wider metastasis has developed. On the other hand, sometimes, unnecessary salvage surgery is performed, which can lead to considerable morbidity. Therefore, there is a need to establish an alternative marker of active disease, for use as a monitoring tool to identify patients with either persistent or emerging disease, to complement and stratify existing clinical and radiological tools.

The majority of LaCC are caused by the human papilloma virus (HPV). During carcinogenesis, the viral DNA that is present in the tumour cell (integrated into the tumour DNA or in episomal form) is released into the blood following tumour lysis and can be measured. Studies have been published demonstrating the utility of circulating tumour DNA as a method of investigating and monitoring tumour biology and clinical status [5,6]; circulating HPV-DNA (cHPV-DNA) can potentially be used as a detection marker for HPV-related LaCC. Published studies have used polymerase chain reaction (PCR) based techniques to detect cHPV-DNA in patients with LaCC at diagnosis; with variable sensitivity of 24–83% [7,8,9,10,11,12,13,14]. The potential of cHPV-DNA as a marker of disease response following radical chemo-radiotherapy in LaCC has not been extensively evaluated.

We developed an ultra-sensitive HPV DNA next generation sequencing (NGS) assay, panHPV-detect, with the ability to comprehensively detect circulating DNA of high-risk HPV genomes (16, 18, 31, 33, 35, 45, 52 and 58) and assess its relationship with disease status and response. We have previously validated this assay in patients with locally advanced anal cancer undergoing CRT. Here, we validate the assay in prospectively collected plasma DNA at serial time points in patients with LaCC treated with primary CRT.

## 2. Materials and Methods

Informed consent was obtained from patients with LaCC planned to receive chemoradiation. The Institutional board (Ref. no. CCR 4157) and ethics committee (Ref. no. 14/NE/1055) approved the study. Treatment consisted of external beam radiotherapy to the pelvis (and para-aortic lymph nodes if common iliac lymph nodes involved) to a dose of 45 Gy in 25 fractions, concomitant cisplatin 40 mg/m^2^ weekly, and image guided adaptive brachytherapy to a combined target dose ≥85 Gy EQD2. Involved lymph nodes were treated to 55–57.5 Gy in 25 fractions using a concomitant boost technique.

Pelvic examination, multiparametric MRI and FDG PET/CT were performed at baseline, end of external beam chemoradiotherapy (week 5) and at 3 months post completion of brachytherapy. In the case of complete local response, further imaging with MRI and CT thorax was deferred until 12 months from completion of treatment, although patients continued to be followed with pelvic examination every 3 months. Standard practice for patients with negative PET/CT but equivocal MRI was to perform repeat MRI 3 months later, i.e., at 6 months. To identify locally persistent or recurrent cervical disease any patients with persistent FDG avidity at any time and corresponding MRI changes underwent examination under anaesthetic and cervical biopsy. Patients with regional or distant metastatic relapse were identified on serial imaging at the timepoints described. 

Serial plasma samples (20 mL) were collected at baseline (before CRT), at week 6–7 (during brachytherapy) and 3 months following completion of treatment in Streck^®^ tubes. Samples of pooled plasma from a set of five healthy pre-screened individuals with no cancer or history of cancer purchased from CTLS (Clinical Trial Laboratory Services), London and 19 patients with breast cancer enrolled in the prospective sample collection study (PlasmaDNA, CCR3297, REC Ref No: 10/H0805/50) were used as negative controls. None of the negative controls were known to have pre-cancerous lesions. Written informed consent was obtained from all control participants.

### 2.1. Blood Samples: Plasma Processing and DNA Extraction

20 mL of blood was centrifuged at 1600× *g* for 10 min within 48 h of collection. The resulting plasma was then centrifuged again at 1600× *g* for 10 min, aliquoted and frozen at −80 °C. DNA was extracted from 5 mL of plasma using the QIAamp Circulating Nucleic Acid Kit (Qiagen, Manchester, UK) according to manufacturer’s instructions. DNA was eluted in 50 μL of AVE buffer and stored at −20 °C. Plasma DNA was quantified using a Bio-Rad QX200 ddPCR system, using ribonuclease P (RNase P) as a reference gene as previously described [15]. 

### 2.2. Tumour Biopsies: HPV DETECTION in Tumour 

Formalin fixed paraffin embedded tumour blocks of the diagnostic biopsy samples were obtained. Eight 10 μm nuclear fast red-stained slides and two haematoxylin and eosin (H&E) stained slides were obtained from representative FFPE blocks. Tumour content, cellularity and suitable areas of tumour were marked for macro-dissection. DNA was extracted using the AllPrep DNA FFPE Kit (Qiagen, Manchester, UK) followed by cDNA synthesis using the Omniscript Reverse Transcription kit (Qiagen, Manchester, UK). 

### 2.3. cHPV-DNA Sequencing in Plasma and Tissue (panHPV-Detect Assay Design)

PanHPV-detect was designed as follows. Representative sub-lineages [16] from each of the eight high risk HPV genotypes associated with 98% of cervical cancers (16, 18, 31, 33, 35, 45, 52 and 58) were aligned using CLUSTAL O (1.2.1) multiple sequence alignment programme (https://www.ebi.ac.uk/ accessed on 30 November 2020). Diagnostic single nucleotide polymorphisms (SNP) were identified for each sub-lineage [17,18,19] and used as a guide for primer design using the Ampliseq Designer (ThermoFisher Scientific, Manchester, UK). Eight primer sets targeting the human genes ACTINB, GAPDH, HPRT were also included in the panel to act as positive controls for library preparation and sequencing efficiency. A total of 2043 SNP targets were submitted; 235 targets were missed thus providing 88.5% coverage. The amplicon length of 140 bp was specified with the primary aim of detection of circulating free DNA fragments in plasma. The panel consisted of 548 primers divided into two pools (Table 1). Ion torrent libraries were prepared using an Ion Ampliseq library preparation kit 2.0 (ThermoFisher Scientific) according to manufacturer’s instructions using 5 ng of tissue DNA or 3 ng of plasma DNA per primer pool. Reads were aligned to an amalgamated reference containing scaffolds for each of the HPV genotypes as well as the reference human targeting genes using TMAP on the Ion Torrent machine. Bedtools v2.23.0 [20] was used to extract on-target reads from the aligned files with a minimum overlap of 90% with amplicons in the panel. Additionally reads with a mapping quality of <30 were removed using samtools v1.2 [21]. Reads were split into those covering human and HPV amplicons and coverage of each portion and each genotype was calculated individually.

We performed an initial validation of panHPV-detect by testing its ability to correctly identify the HPV sub-type in cervical cancer tissue samples. We obtained tumour DNA extracted from cervical cancer tissue previously typed for the HPV sub-type using polymerase chain reaction (RT-PCR) for E6 and E7 mRNA using validated assays. Five samples for each of the eight high-risk HPV sub-types were obtained from the Scottish HPV archive. Samples from five HPV negative patients, with head and neck cancer recruited in the head and neck cancer cohort of this study and 19 patients with breast cancer enrolled in the prospective sample collection study (PlasmaDNA, CCR3297, REC Ref No: 10/H0805/50) were used as negative controls. HPV status was confirmed as negative with E7 RT-PCR performed on RNA extracted from the samples. None of the negative controls were known to have pre-cancerous lesions. Written informed consent was obtained from all participants.

Following the initial validation, we investigated whether panHPV-detect was able to detect cHPV-DNA in pre-treatment plasma of patients recruited in the study. We then tested the concordance of HPV subtype identified by panHPV-detect in plasma with the HPV subtype identified in the tumour biopsies collected from the patients at diagnosis. 

### 2.4. Sample Size Calculation

The primary endpoint for this pilot study was the feasibility of detecting cHPV-DNA at baseline in patients undergoing CRT with radical intent. Our pilot study in HPV+ head and neck cancer demonstrated 100% sensitivity and 93% specificity in detecting cHPV-DNA in patients’ plasma at baseline (pre-treatment) using the original NGS assay (HPV-detect) [6]. To prove 85% sensitivity assuming that the true sensitivity is 99% and to have 80% power given two-sided type I error of 0.05, would require at least 19 HPV+ patients, all of them expected to be labelled as positive by NGS. Since ~85% of LaCC patients are HPV+, therefore, the study required 22 patients. The secondary endpoint was to assess the potential of cHPV-DNA to predict response to treatment.

### 2.5. Data Analyses

Sensitivity and Specificity were calculated using contingency tables. These were calculated with the tissue HPV status and sub-type obtained from the patient’s tumour biopsy as the gold standard test. All statistical analyses were performed using GraphPad Prism version 7 software. 

To classify HPV+ and HPV− samples using panHPV-detect in tissue, we set a threshold whereby a sample was classified positive if there were ten reads present from more than 30% of the different HPV amplicons for each sub-type. To assess the threshold for the number of amplicons needed for positive panHPV-detect readout in plasma—a ROC analysis was used. In the first step, HPV status and subtype was assigned in tissue using E7 mRNA to separate the two groups. The number of amplicons with greater than ten reads at baseline was input for each patient to find a suitable threshold for this parameter. Sorting the values in both HPV+ and negative groups and averaging adjacent values in the sorted list generated a list of thresholds. Based on the ROC analysis, a threshold that gave the greatest sensitivity and specificity for each HPV subtype was selected as the threshold for classification of plasma as HPV DNA positive. Based on the ROC analysis, a threshold of 6.5 amplicons with more than ten reads and 7.5 amplicons in more than ten reads was set for HPV16 and HPV18, respectively. These were the thresholds that gave the greatest sensitivity and specificity and were selected as thresholds for classification of plasma as HPV DNA positive. 

There were insufficient plasma samples for the other rarer HPV subtypes to provide a statistically robust threshold and therefore, an empirical and pragmatic (between 6.5 and 7.5 amplicons) threshold of seven amplicons with greater than ten reads were set for the other subtypes. 

## 3. Results

Tumour DNA from 39/40 pre-typed tumour samples (five of each of the HPV sub-types 16, 18, 31, 33, 35, 45, 52 and 58) obtained for validation was of satisfactory quality for the initial tissue validation. PanHPV-detect demonstrated 100% sensitivity and specificity in correctly identifying the HPV sub-type in tumour DNA and negative controls (Table 2). 

Twenty-two patients were recruited into the study. The median age was 48 (range 33–88). Patient characteristics are described in Table 3. Tumour tissue and baseline blood samples were available for 20 patients. In two patients’ plasma samples were of poor quality and unsuitable for analysis. Tumour tissue in 18/20 patients was positive for HPV. cHPV-DNA was detected by panHPV-detect in 16/18 plasma samples. The tissue sample from one patient was positive for HPV59, which was not included in panHPV-detect. cHPV-DNA was not detected in plasma of HPV negative patients. Therefore, panHPV-detect demonstrated a sensitivity and specificity of 88% (95% CI-70–99%) and 100% (95% CI-30–100%), respectively. 

In 16 samples with detectable cHPV-DNA, the following sub-types were correctly identified as follows—HPV16—11 samples, HPV18—5 samples, HPV45—2 samples and HPV58—3 samples. Some samples had multiple HPV-subtypes. See Table 4. 

The median follow-up for all patients was 16 months (range 11–29). A total of 16 of the 22 patients recruited in the study had a complete response (CR) and six patients had equivocal disease according to MRI and PET-CT assessment at 3 months following completion of CRT. Of the 16 patients with detectable cHPV-DNA at baseline, cHPV-DNA was detected in plasma at 3 months in the two patients with residual disease and one patient with clinical CR. Of the two patients with detectable cHPV-DNA and radiological residual disease, one had disease in the para-aortic nodes and the other had residual disease in the cervix. The para-aortic disease was treated with radiotherapy. The other patient was referred for salvage surgery; however, this was not undertaken due to significant comorbidities. The patient with detectable cHPV-DNA and clinical CR at the 3-month time-point had clinical relapse at 14 months post-treatment. This was treated with salvage surgery. Four patients with radiological partial or equivocal response and undetectable cHPV-DNA at the 3-month time point and eight patients with CR and undetectable cHPV-DNA at the 3-month time-point were disease free at the last follow-up appointment. One patient with CR and undetectable cHPV-DNA at the 3-month time-point relapsed at 14 months post-CRT. This was successfully treated with salvage surgery. Of the 16 patients with detectable cHPV-DNA at baseline, panHPV-detect accurately predicted presence/absence of residual disease (macroscopic/microscopic) in 15 out of the 16 patients. 

## 4. Discussion

We have designed “panHPV-detect” a novel NGS based method for detection and tracking of cHPV-DNA from eight high-risk HPV genotypes. The assay was initially tested in banked HPV-typed cervical tissue samples. Following this, we tested this assay in a prospective observational biological sample collection study in patients undergoing radical CRT and we have demonstrated high sensitivity and specificity. Tracking cHPV-DNA in sequential samples through and after CRT accurately predicted response and residual disease suggesting the potential of panHPV-detect to enhance clinical decision-making.

Compared to PCR-based assays, amplicon-based NGS of multiple regions of the viral genome is able to detect cHPV-DNA at baseline, with higher sensitivity and specificity. To our knowledge this is the first study to use NGS to detect cHPV-DNA in patients undergoing CRT for LaCC. Previously published studies used PCR for cHPV-DNA detection, with reported sensitivity of detecting cHPV-DNA at baseline of between 24–83% [7,8,9,10,11,12,13,14] (Table 5). The majority of these studies used primers specific to HPV16 and 18. Although, 16 and 18 are the oncogenic HPV subtypes in approximately 80% of the HPV associated LaCC, excluding the other high-risk sub-types reduces the sensitivity of the assay. Despite using an identical PCR primer set for the seven commonest high-risk sub-types, two studies demonstrate sensitivity of 12% and 65% [9,11]. Our NGS based assay which is able to detect multiple HPV genomic regions across the eight most common high-risk sub-types, demonstrates a superior sensitivity to the PCR based technique.

We envisage several uses for cHPV-DNA in LaCC. Firstly, as an adjunct to standard-of-care MRI and/or PET-CT scans, cHPV-DNA could potentially confirm complete response at 3 months, precluding the requirement for repeated scans and biopsies. This would result in a significant health economic impact and avoid unnecessary patient morbidity and anxiety. 

This is supported in our study by the observation that 14 out of 16 patients with detectable baseline cHPV-DNA in our study had undetectable cHPV-DNA at 3 months post-CRT. Out of the 14, four had partial or equivocal radiological response; however, no evidence of disease relapse was observed on subsequent imaging and these patients remain relapse free at the last follow-up. 

Secondly, in other cases of equivocal imaging and where biopsy is either not possible or negative, cHPV-DNA could be used to confirm both local and metastatic disease. Three patients in our study had detectable cHPV-DNA at 3 months and subsequently had confirmed relapse. 

To our knowledge only three other studies correlated cHPV-DNA levels to disease response. In the study by Campitelli et al. [13] high cHPV-DNA levels in 2 patients (out of their 16 patients with post-CRT blood samples) correlated with disease relapse and in the study by Yang et al. [10] 5 patients (out of their 21 patients with post-CRT blood samples) had detectable cHPV-DNA all of whom had disease relapse. Elevated cHPV-DNA levels at 3-months following CRT in our study and the ones by Campitelli et al. [13] and Yang et al. [10] accurately predict residual disease and/or disease relapse (10/53 patients with detectable cHPV-DNA at 3 months, the three studies combined). More recently Leung et al. [22] evaluated an NGS technique with 100% sensitivity and 67% specificity for detection of recurrence. Patients considered to have residual or recurrent cervical disease may be evaluated for salvage surgery. Although repeat cervical biopsy is undertaken prior to surgery, in the post CRT setting this is not targeted and difficult to interpret. Some patients will therefore proceed to surgery without positive biopsy and ultimately will not have pathological evidence of disease on hysterectomy specimen. Surgical complications of wound infection and healing are increased after CRT. 

Another role for cHPV-DNA is in the monitoring or follow up phase. Patients with persistent cHPV-DNA at 3 months and negative imaging could be triaged to more intensive imaging follow up, hopefully to capture early emergence of relapse sites potentially amenable to targeted treatment. Furthermore, using more sensitive assays such as plasma cHPV-DNA as a marker for detection of true high-risk patients could aid the design of adjuvant therapy studies. 

Using panHPV-detect we were able to accurately identify the specific sub-type of the cHPV-DNA, which possibly indicates the causative oncogenic sub-type. Studies have suggested that the causative HPV sub-type can affect prognosis with the alpha-7 (HPV18, 39, 45) sub-types having inferior outcomes compared to alpha-9 (HPV16, 31, 33, 52, 58) sub-types following CRT [23] and worse prognosis following primary surgery in cervical cancers related to HPV 18 [24]. Therefore, identification of the correct sub-type driving the oncogenic process may provide prognostic information. 

This was a pilot study to examine the potential of a novel NGS assay to detect cHPV-DNA before radical therapy and identify the causative HPV sub-type. This study did not have adequate power to so. However, it confirms the findings from studies by Campitelli et al. [13] and Yang et al. [10] highlighting the potential of cHPV-DNA in predicting disease response following CRT. 

## 5. Conclusions

cHPV-DNA has significant potential as a biomarker of response following radical intent CRT and of recurrence during surveillance for patients with LaCC. Furthermore, panHPV-detect can identify cHPV-DNA with high sensitivity and specificity enabling the use of cHPV-DNA as a biomarker to become a reality, albeit further validation in larger multi-centre studies is required. Such validation studies of panHPV-detect in patients undergoing curative intent CRT in LaCC are currently in development.

## Figures and Tables

**Table 1 cancers-15-01387-t001:** Details of panHPV-detect panel primer pools.

HPV	Primers in Pool 1	Primers in Pool 2	Total
HPV16	40	39	79
HPV18	41	41	82
HPV31	33	32	65
HPV33	35	35	70
HPV35	15	14	29
HPV45	34	34	68
HPV52	38	38	76
HVP58	36	35	71

**Table 2 cancers-15-01387-t002:** Sensitivity and specificity of panHPV-detect assay following initial validation of panHPV-detect in cervical cancer tissue samples obtained from another biobank that were confirmed HPV positive. Refer to text for details on negative controls.

		Tissue HPV Status—PCR		
panHPV-detect		Positive	Negative	Total
	Present	39	0	
	Absent	0	24	
	Total	39 (Sensitivity 100%)	24 (Specificity 100%)	63

**Table 3 cancers-15-01387-t003:** Patient characteristics for the cohort.

Variable			Number of Cases
Number of patients			22
FIGO Stage	I		
		A	0
		B	2
	II		
		A	2
		B	15
	III		
		A	0
		B	1
	IV	C	0
		A	1
		B	0
Pack-years (Includes ex Smokers)	NS (Never Smoked)		15
	≤20		4
	>20		3

**Table 4 cancers-15-01387-t004:** Details the HPV status in tissue at diagnostic pathology and in tissue and plasma using panHPV-detect.

Patient Trial ID	HPV-Status Tissue Diagnostic Pathology	HPV-Status panHPV-Detect Tissue	HPV-Status panHPV-Detect Plasma
CCR 4157 (C01)	18	18	18
CCR 4157 (C02)	16	16	16
CCR 4157 (C03)	16	16	16
CCR 4157 (C04)	16	16	16
CCR 4157 (C05)	neg	neg	neg
CCR 4157 (C06)	neg	neg	neg
CCR 4157 (C07)	16	16	16
CCR 4157 (C08)	16	tissue block not available	16
CCR 4157 (C09)	18	18	not detected
CCR 4157 (C010)	45	45	45
CCR 4157 (C011)	unknown	16,58	16, 58
CCR 4157 (C012)	not detected	18	18
CCR 4157 (C013)	59	not in panel	NA
CCR 4157 (C014)	unknown	16,58	16, 58
CCR 4157 (C015)	16 D	16	no BL blood
CCR 4157 (C016)	18	16,18	16, 18
CCR 4157 (C017)	not done	16,18	16, 18
CCR 4157 (C018)	18	18	no BL blood
CCR 4157 (C019)	18	no block	18
CCR 4157 (C020)	unknown	16,58	16, 58
CCR 4157 (C021)	18	16,18	16, 18
CCR 4157 (C022)	45	45	45

**Table 5 cancers-15-01387-t005:** Studies reporting PCR techniques for HPV DNA detection.

Author	Technique	Stage	Sample Size	Sensitivity	Specificity	HPV Sub-Types	Serial Samples
Cheung [7]	PCR	Non-metastatic	138	55.8%	NR	16, 18	No
Hsu [8]	PCR	Non-metastatic	112	24.1%	100%	16, 18	No
Satish [9]	PCR	Non-metastatic	58	11.8%	100%	16, 18, 31, 33, 35, 45, 58	No
Yang [10]	PCR	Non-metastatic	50	50%	85%	16, 18	Yes
Wei [11]	PCR	Non-metastatic	23	65%	100%	16, 18, 31, 33, 35, 45, 58	No
Jaberipour [12]	PCR	Metastatic and Non-metastatic	69	23.5%	90%	16, 18, 33, 52	No
Campitelli [13]	PCR	Metastatic and Non-metastatic	16	81.25%	100%	16, 18	Yes
Jeannot [14]	PCR	Metastatic and Non-metastatic	47	83%	100%	16, 18	No
Current study	NGS	Non-metastatic	22	88%	100%	16, 18, 31, 33, 35, 45, 52 and 58	Yes

## Data Availability

Data are contained within the article.

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
