# Peer review of "Predicting Response to Radical Chemoradiotherapy with Circulating HPV DNA (cHPV-DNA) in Locally Advanced Uterine Cervix Cancer"

_cancers, 2023, doi:10.3390/cancers15051387_

Round 1

Reviewer 1 Report

The manuscript does not show a lot of data. It only shows the interpretation of this data in a single table. This leaves the reader in doubt. The authors should show more data to support their inferences, as indicated in my comments.

Author Response

Referee #1 (Comments to the Author):

The authors would like to thank the reviewer for the positive comments about our study.

Comments - The manuscript does not show a lot of data. It only shows the interpretation of this data in a single table. This leaves the reader in doubt. The authors should show more data to support their inferences, as indicated in my comments.

Response- This has now been actioned and more detailed data has been provided with regards to panel design, patient characteristics and sample analyses. We trust that this will be satisfactory.

Reviewer 2 Report

In this manuscript, Susan et al., developed a pan-HPV detect test kit and demonstrated impressive detection sensitivity and specificity for LaCC cancer. These findings may provide important clues for the early detection of LaCC cancer. The paper was written in a clear and logical manner, and the analysis and discussion are thorough. However, from the reviewer's personal point of view, there are 2 concerns need to be addressed before this manuscript can be considered for acceptance.

1, Please provide more information for the panHPV-detect design and the NGS assay. (e.g, PCR primer sequence, number of NGS reads, aligned reads, total reads required per gene/sample )

2, please provide more information about the selection threashold and ROC analyses.

Author Response

The authors would like to thank the reviewer for the positive comments about our study.

In this manuscript, Susan et al., developed a pan-HPV detect test kit and demonstrated impressive detection sensitivity and specificity for LaCC cancer. These findings may provide important clues for the early detection of LaCC cancer. The paper was written in a clear and logical manner, and the analysis and discussion are thorough. However, from the reviewer's personal point of view, there are 2 concerns need to be addressed before this manuscript can be considered for acceptance.

Comments

  1. Please provide more information for the panHPV-detect design and the NGS assay. (e.g, PCR primer sequence, number of NGS reads, aligned reads, total reads required per gene/sample )

Response- As requested further details have been provided as regards the pnHPV-detect design in the methods section, Given that the panel was designed using a commercial partner- AmpliSeq (ThermoFisher Scientific), we were not provided with the primer pool sequences as these have a proprietary protection. We ordered the designed primer pools from AmliSeq as required for analysis.

  1. please provide more information about the selection threshold and ROC analyses.

Response – This has now been detailed in the data analyses section.

Reviewer 3 Report

HPV is considered as high risk oncovirus in Cervical cancer, Head and neck cancer. Predicting response to radical chemoradiotherapy with circulating HPV DNA (cHPV-DNA) in locally advanced uterine cervix cancer would be interesting.

However, this study did not provide enough data.

Major:

1. Please provide raw data, other than a simple table.

2. Please note what exact statitistic methods you used, but not Graphpad.

3. You should provide evidence to support the samples you utilized are HPV+ or HPV- as what you claimed.

4. Overall, no direct supporting data, how do people trust what you claim only in words and translated numbers.

Author Response

Reviewer 3

HPV is considered as high risk oncovirus in Cervical cancer, Head and neck cancer. Predicting response to radical chemoradiotherapy with circulating HPV DNA (cHPV-DNA) in locally advanced uterine cervix cancer would be interesting.

The authors would like to thank the reviewer for the positive comments about our study.

However, this study did not provide enough data.

Comments-

Major:

  1. Please provide raw data, other than a simple table.

Response- This has now been actioned and more detailed data has been provided with regards to panel design, patient characteristics and sample analyses. We trust that this will be satisfactory.

  1. Please note what exact statitistic methods you used, but not Graphpad.

These have been detailed in paragraph 1- data analyses section

  1. You should provide evidence to support the samples you utilized are HPV+ or HPV- as what you claimed.

The diagnostic tissue samples were typed for HPV in the Institutes pathology laboratory using the standard validated assays that are in use in the clinical practice. Furthermore, HPV status and sub-type were also confirmed using panHPV-detect. Details provided in table 4.  

  1. Overall, no direct supporting data, how do people trust what you claim only in words and translated numbers.

Response: We trust that this more detailed data now provided will rectify this aspect of the original submission.

Round 2

Reviewer 3 Report

This manuscript has been improved a lot, however, my previous questions were not answered.

Add two more specific questions:

1. What are the exact primer sequences for HPVs?

2. What exact statistical methods used in the statistical analyses. All the same, or they were differentially used.

Author Response

What are the exact primer sequences for HPVs?

Given that the panel was designed using a commercial partner- AmpliSeq (ThermoFisher Scientific), we were not provided with the exact primer pool sequences as these have a proprietary protection. We ordered the designed primer pools from AmliSeq as required for analysis.

What exact statistical methods used in the statistical analyses. All the same, or they were differentially used

These have now been further clarified in These have been detailed in paragraph 1- data analyses section